

# Thwaites Eastern Ice Shelf Cavity Observations Reveal Multi-year Sea Ice Dynamics and Deep-Water Warming in Pine Island Bay, West Antarctica

Christian T. Wild[1,2], Tasha Snow[3,4,5], Tiago S. Dotto[6], Peter E.D. Davis[7], Scott Tyler[8], Ted A. Scambos[9], Erin C. Pettit[2], Karen J. Heywood[10]

[1]Department of Geosciences, University of Tübingen, Tübingen, Germany
[2]College of Earth, Ocean, and Atmospheric Sciences, Oregon State University, Corvallis, OR, USA
[3]University of Maryland, ESSIC, College Park, MD, USA
[4]NASA Goddard Space Flight Center, Greenbelt, MD, USA
[5]Colorado School of Mines, Golden, CO, USA
[6]National Oceanography Centre, Southampton, UK
[7]British Antarctic Survey, Natural Environment Research Council, Cambridge, UK
[8]University of Nevada, Reno, USA
[9]Earth Science and Observation Center, CIRES, University of Colorado Boulder, Boulder, CO, USA
[10]Centre for Ocean and Atmospheric Sciences, School of Environmental Sciences, University of East Anglia, Norwich, UK

*Correspondence to*: Christian T. Wild (christian.wild@uni-tuebingen.de)

**Abstract.** Pine Island Bay, situated in the Amundsen Sea, is renowned for its retreating ice shelves and sea ice variability. Brine rejection from sea ice formation and glacial meltwater exported from ice-shelf cavities impact seawater density and thus regional ocean circulation. While the effects of brine rejection on the continental shelf are relatively well documented, little is known about its effects on water subsequently circulating beneath floating ice shelves. Here, we present insights from oceanographic instruments deployed via boreholes into the ocean cavity beneath the Thwaites Eastern Ice Shelf (TEIS) from 2020 to 2023. These observations reveal warming and thickening of the modified Circumpolar Deep Water (mCDW) layer near the seabed since January 2020. Concurrently, multi-year sea ice anchored along the coastline has retreated over 150 km to the calving fronts of Pine Island and Thwaites Glaciers, leading to increased Winter Water advection and a cooling of over 1˚C in the upper 250 m below TEIS between July 2021 and January 2023. The causal link between sea ice dynamics and changing hydrographic properties in the subshelf cavity is supported by distinct events lasting several weeks during periods of mobile sea ice coverage. During these events, mid-depth waters temporarily warm and increase in salinity, leading to an increase in density, while deeper mCDW simultaneously cools and becomes fresher, reducing its density. These observations are important for refining process models and enhancing the accuracy of basal melt-rate parametrizations for coupled ice-ocean modelling.



## 1 Introduction

Ice shelves encircle much of Antarctica, acting as critical buffers that slow the flow of continental ice into the ocean (Fürst et al., 2016). However, many ice shelves have thinned or even collapsed in recent decades (Doake and Vaughan, 1991; Rack and Rott, 2004; Scambos et al., 2004; Lhermitte et al., 2023), triggering rapid acceleration of grounded ice (Rignot et al., 2004; Scambos et al., 2014). This process is particularly concerning in the Amundsen Sea Embayment, where Pine Island and Thwaites Glaciers could contribute together 1.16 m to global sea-level rise if marine ice-sheet instability takes hold (Schoof, 2007; Joughin et al., 2014; Rignot et al., 2019; Gudmundsson et al., 2023; Morlighem et al., 2024). Thwaites Glacier has become a focal point in climate research (Scambos et al., 2017) due to its rapid retreat (Rignot et al., 2019; Milillo et al., 2019; Wild et al., 2022; Rignot et al., 2024) and the on-going deterioration of its last remaining ice shelf (Alley et al., 2021; Wild et al., 2024), largely driven by the intrusion of modified Circumpolar Deep Water (mCDW; Dutrieux et al., 2014; Christianson et al., 2016; Jenkins et al., 2018; Nakayama et al., 2019). However, sub ice-shelf cavities remain among Earth's least explored regions, and limited observational data hinder our ability to model the intricate interplay between oceanic warming, ice-shelf stability, grounding-zone processes, and the fate of Thwaites Glacier (Seroussi et al., 2017; Yu et al., 2018; Holland et al., 2023).

Circumpolar Deep Water accesses the continental shelf through deep glacially carved troughs (Heywood et al., 2016). It gradually cools and freshens as it moves southward, following narrow bathymetric pathways (10–20 km wide) and mixing with on-shelf water masses before intruding into the deeper cavities beneath ice shelves and glacier fronts (Nakayama et al., 2019). By the time it reaches Pine Island Bay (PIB), mCDW (>0 °C, >34.7 g kg$^{-1}$) remains 2–4 °C above the in-situ freezing point, supplying substantial thermal energy for basal melting. The Thwaites Trough extends from the north, reaching depths of ~1300 m and splitting into three narrower branches west of the pinning point buttressing Thwaites Eastern Ice Shelf (TEIS), while the adjacent Pine Island Bay Trough, slightly deeper (~1400 m), extends beneath TEIS from the east but is thought to be constrained by a bathymetric sill (Fig. 1a). Autonomous underwater vehicle (AUV) surveys indicate that mCDW enters the TEIS cavity predominantly through the easternmost branch near its pinning point (T3), with meltwater-enriched waters exiting through the westernmost branch (T2, Fig. 1a; Wåhlin et al., 2021). Notably, hydrographic signatures from PIB have been detected near the pinning point (Biddle et al., 2019), suggesting mixing between these two competing water masses at depth and an extensive westward influence of PIB circulation (Seroussi et al., 2017; Nakayama et al., 2019).




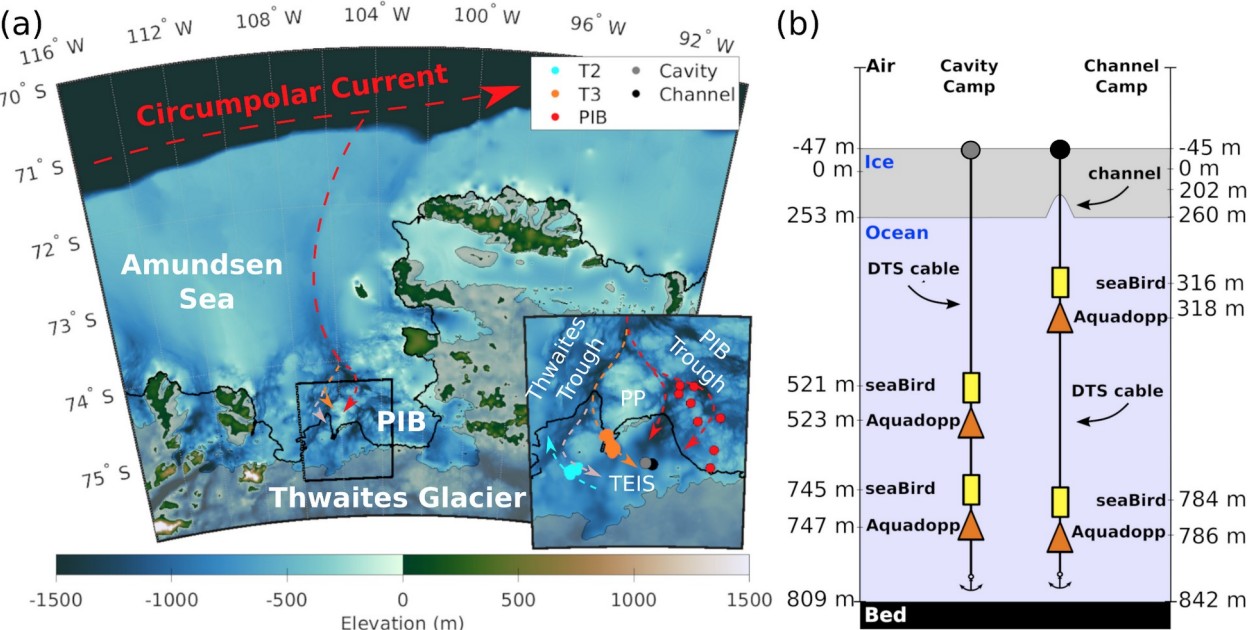

**Figure 1:** (a) Bathymetric map showing water pathways into Pine Island Bay (PIB). The inset shows the location of Cavity Camp and Channel Camp on Thwaites Eastern Ice Shelf (TEIS) and the location of its pinning point (PP). Red dots indicate locations of ship-based CTD measurements capturing PIB water masses, while light blue and orange dots represent AUV measurements in the bathymetric troughs T2 and T3, respectively, which branch from the Thwaites Trough (Wåhlin et al., 2021). (b) Illustration presenting a cross-sectional view of an idealized ice-shelf featuring a basal channel, showing the positions of Cavity Camp and Channel Camp, the two DTS cables, MicroCATs, and Aquadopp instrument pairs deployed in the subshelf ocean cavity.

Observational studies have demonstrated that subshelf oceanography is strongly influenced by neighbouring ocean conditions (Webber et al., 2017; Dotto et al., 2022, Davis et al., 2023). AUV and ship-based Conductivity-Temperature-Depth (CTD) surveys have revealed competing mCDW sources beneath TEIS, originating from both PIB and Thwaites Trough (Wåhlin et al., 2021). In PIB, surface circulation is dominated by a gyre system—a rotating ocean circulation shaped by regional wind forcing, bathymetry, and glacial meltwater fluxes (Thurnherr et al., 2014; Heywood et al., 2016; Yoon et al., 2022). Its strength and sense of rotation can be altered by the concentration and mobility of landfast sea ice—stationary, often multi-year sea ice anchored to the coastline (hereafter, 'fast ice') that eventually forms a stable, immobile platform that isolates the ocean from atmospheric wind stress (Zheng et al., 2022). Extended periods of fast ice coverage promote weakening of the PIB gyre leading to an accumulation of glacial meltwater (i.e., a relatively warmer water derived from mCDW melting the ice base) near the surface, which leads to shallower isopycnals beneath the neighbouring TEIS and thus to warmer conditions at the TEIS base (Dotto et al., 2022). In contrast, fast ice breakouts combined with a cyclonic PIB gyre enhance the intrusion of cooler surface waters into the subshelf cavity (Dotto et al., 2022), potentially explaining the suppressed basal melt beneath the ice shelf (Wild et al., 2024).



Previous observations have provided valuable insights, but their spatio-temporal limitations fail to capture the multi-year variability in how fast ice influences hydrographic properties. In particular, the role of different sea ice types in modulating ocean surface stress and gyre strength can be significant (Zheng et al., 2022), and this process may directly impact the heat available for basal melting beneath nearby ice shelves (St-Laurent et al., 2015). Furthermore, the vertical extent of warmer conditions in subshelf cavities identified by Dotto et al. (2022) during extended fast ice coverage remains unknown. Here, we extend the observational record presented by Dotto et al. (2022) from January 2020–March 2021 to January 2020–January 2023 to investigate how the formation of thin and mobile, first-year sea ice contrasts with thick and immobile, multi-year fast ice in shaping ocean conditions beneath TEIS. Additionally, we assess how the competing water masses from PIB and Thwaites Trough respond to the persistence and extent of multi-year fast ice.

The paper is organized as follows: First, we present the dataset and analyze the temporal variability of hydrographic properties at shallow, mid-depth, and deep water layers. Next, we compare our measurements beneath TEIS with published datasets from nearby ship-based surveys. We then examine the temporal co-variability of our expanded dataset, revealing a progressive warming of the mCDW layer at depth, periodically disrupted by distinct events lasting a few weeks in which the mCDW temporarily cools and freshens, while mid-depth waters become denser. Using Distributed Temperature Sensing (DTS) profiles, we assess the vertical extent of these events throughout the water column. Finally, we analyze remotely sensed sea ice cover in PIB, identifying that events align with first-year sea ice formation that persist until May 2021. After this period, the upper water column undergoes substantial cooling, likely driven by the gradual retreat of multi-year fast ice in PIB. This retreat enhances Winter Water (WW) formation through air-sea fluxes (Webber et al., 2017), promoting the intrusion of WW beneath the adjacent TEIS.

## 2 Data and methods

### 2.1 Observations and processing

In December 2019, we established two hot water drilling camps on TEIS to access its underlying ice-shelf cavity: Cavity Camp, situated centrally above the ocean cavity beneath the ice, and Channel Camp, positioned at the apex of an ice-shelf basal channel (Fig. 1; Dotto et al., 2022; Scambos et al., 2025). We present atmospheric and hydrographic measurements of both sites collected between January 2020 and January 2023 by two automated stations (Automated Meteorology-Ice-Geophysics Observing Systems - 3, or AMIGOS-3; Scambos et al., 2025). These on-ice mooring systems incorporated instruments on the ice-shelf surface (e.g., air temperature, wind, and pressure sensors), and DTS fiber optic systems drilled through the ice shelf and the entire water column beneath to capture ice and ocean temperature profiles. Each AMIGOS-3 station also included an under-ice mooring with a suite of ocean instruments attached (described in detail below), including a set of MicroCAT instruments for measuring ocean conductivity, temperature, and pressure, each paired with Aquadopp current meter instruments (Fig. 1b).



### 2.1.1 Atmospheric dataset

We used wind speed and direction measurements to determine the prevailing atmospheric circulation that may impact ice and ocean processes near TEIS. The AMIGOS-3 were equipped with a multi-parameter Vaisala 530 series weather sensor, which acquired hourly air temperature, wind speed and direction at 7 to 3 m above the surface of the ice shelf (as accumulation slowly buried the AMIGOS-3 tower). Here we focussed on the atmospheric data record from Channel Camp as the difference in atmospheric variability from Cavity Camp is negligible within the context of this study, and the Channel

Camp data record is slightly longer (Scambos et al., 2025). Given the potential influence of atmospheric winds on upper ocean circulation patterns, we compared the wind data with the variability observed in ocean sensors measuring current speed and direction. For this comparison we relied on ERA5 reanalysis on single levels (Hersbach et al., 2020) because of temporal gaps in our wind record, which were caused by rime and heavy snowfall on the sensor (April 19–May 19, 2020; June 30–July 23, 2020; and August 8–September 11, 2020). From ERA5's 0.25° × 0.25° spatial resolution, we selected and

averaged three grid points (Latitude: -75°, Longitudes: -105.76°, -105.51°, and -105.26°) to obtain a representative dataset for the TEIS region. We used ERA5's native hourly resolution for wind speed, wind direction, and 2 m temperature, and subsequently averaged the atmospheric dataset into daily bins. The validity of ERA5 was assessed by comparing it to our wind measurements during periods when observations were available (Appendix A).

### 2.1.2 Borehole CTD cast

On January 12, 2020, hot water drilling activities were conducted at Channel Camp, followed by the collection of an initial CTD profile down to the seabed at a depth of 842 m. This initial CTD cast was used to establish the relationship between temperature, salinity, and ambient pressure within the ocean cavity (Appendix B). To focus on long-term averages we excluded the depth range of the thermocline, between 270 m and 425 m, and fitted a second-order polynomial function to the remaining CTD measurements.

### 2.1.3 MicroCAT CTDs


Four Sea-Bird MicroCAT SBE 37-IMP instruments were employed in fixed depths to monitor temporal variability of conductivity, temperature, and ambient pressure in three distinct water layers. One was positioned at an initial depth of 316 m (referred to as the "shallow" MicroCAT), while a second one was positioned at 521 m ("mid-depth" MicroCAT), and two other sensors were positioned at 745 m and 784 m depth ("deep" MicroCATs) beneath the ocean surface (Fig. 1b). We

conducted cross-calibration of these instruments in the circulating seawater tanks at McMurdo Station.

Following two years of uninterrupted recording at a temporal resolution of 10 minutes, the shallow MicroCAT instrument stopped functioning in January 2022. The mid-depth and both deep MicroCAT instruments remained operational for an additional year until January 2023, when the dataset was retrieved from the instruments. Conservative temperature ($\Theta$; ˚C), absolute salinity ($S_A$; g kg$^{-1}$) and potential density referenced to zero pressure from each instrument were computed





using the Thermodynamic Equations of Seawater-10 (McDougall et al., 2011). We then used a Chebyshev low-pass filter with a 1 hour cutoff frequency to filter these records for outliers and calculated depth below the ocean surface from the filtered in-situ pressure measurements.

### 2.1.4 DTS thermal profiling

DTS temperature profiles through the ocean column were used as a proxy for hydrographic variability at different depths and 145 over varying time scales. A DTS laser interrogator system (Silixa XT, Silixa LTD, Hertfordshire UK) was attached to an armored multi-strand, fiber-optic cable (FIMT) connected to the primary steel cable holding the ocean instruments (Scambos et al., 2025). This setup enabled the collection of temperature profiles with a vertical sampling of 25 centimeters resulting in an approximate spatial resolution of 50 cm (Tyler et al., 2009). DTS measurements were integrated over 1 minute with estimated temperature resolution of 0.033 ˚C and 0.038 ˚C at the deepest measurement for Cavity and Channel Camp 150 mooring, respectively. The temperature resolution is estimated by calculating the variance of DTS-derived temperatures within a 2.5 m section near the bottom of each mooring. The 2.5 m sections were centered at 730 m for Cavity Camp and 750 m for Channel Camp, deep in the profile where no vertical gradients would be measurable over the 2.5 m section.

DTS measurements at both stations were generally captured every 4 hours during the austral spring to early-autumn (October-April), but were extended to 24-hour intervals from mid-autumn through winter (May-September) to conserve 155 power. At Channel Camp, DTS data were acquired from January 2020 to August 2021. In January 2023, we gathered additional DTS data at Channel Camp, with recordings every ~90 seconds over a duration of 2 hours and 45 minutes (UTC Start: January 8, 2023 21:31:37; End: January 9, 2023 00:15:53). Subsequently, these 154 individual DTS profiles from that short period were averaged to create a consolidated DTS profile for January 2023. At Cavity Camp, the DTS data record spans January 2020 until October 2021.

We calibrated the DTS data using the MicroCAT instruments, which sampled the water column during the DTS measurements. For most of the record, we applied a straightforward two-point calibration (slope and offset) to each DTS trace. In 2023, when only the deep MicroCAT instrument was operational at Channel Camp, we performed a three-point calibration using an assumed constant minimum ice temperature from the middle of the ice shelf layer and the pressure melting point at the ice shelf-ocean interface. In both cases, we used a single-ended calibration method. The calibrated DTS 165 data were binned into daily bins.

After calibration, we used the relationship between in-situ temperature and salinity from the initial CTD cast to calculate Θ profiles based on the 'proxy salinity profiles'. This approach assumes that the proxy salinity profile derived on January 12, 2020, remains representative throughout the three years of DTS data collection. To validate this assumption, we compared it against a time series of in-situ temperature, salinity, and pressure from two MicroCATs. We calculated Θ in two 170 ways: (1) using the salinity time series and (2) using a constant salinity from the initial measurement. The differences between these two methods were negligible (RMSE of 0.0002 °C for the shallow MicroCAT and 0.001 °C for the deep MicroCAT at Channel Camp, compared to mean values of -0.88 °C and 1.05 °C, respectively). Based on these results and





the lack of other measurements, we assume a constant salinity profile to derive seawater density profiles, allowing us to assess the net effect of in-situ temperature changes on mean water-column density (Appendix B). A caveat of this
assumption is that this approach primarily captures warm/salty and cold/fresh water masses, and does not account for the warm/fresh combination typical for glacial meltwater.

### 2.1.5 Aquadopp current meters

Nortek Aquadopp current meters were installed two meters below each MicroCAT, capturing current velocities to determine the ocean circulation patterns related to the water characteristics captured by the CTD and DTS systems (Fig. 1b). Aquadopp
data were uplinked by the inductive modem and Iridium data transmission only (Aquadopp systems with internal inductive modems only expose the most recent 48 measurements to the inductive modem via a ring-buffer). Ocean current data were acquired hourly with a data gap between August 10-28, 2020, for the Channel mooring, and May 29 to August 28, 2020, for the Cavity mooring, owing to low station power. The velocity components measured by the Aquadopps were corrected for the magnetic declination, 50.07˚E. The Aquadopp records were binned into daily data chunks to show temporal variability of
ocean current speed and direction.

### 2.2 Monitoring sea ice variability remotely

### 2.2.1 Satellite SAR data from Sentinel-1A

Since water circulation beneath TEIS is likely to be impacted by regional sea ice coverage (Dotto et al., 2022), we used publicly available satellite radar imagery from the Sentinel-1A operating at C-band (5.4 GHz/5.6 cm) to monitor sea ice
variability in PIB. This active microwave sensor has captured synthetic aperture radar (SAR) images every 12 days over PIB since 2014, having the advantage of being able to continuously observe the surface in polar night and through cloud cover, unlike optical imaging systems. We used the extra wide swath mode product with single HH (i.e., horizontally transmitted and horizontally received radar signals) polarization, covering a broad 400 km area at a medium ground resolution of 20 m by 40 m. Using these images, we compiled a video illustrating the regional evolution of sea ice in PIB (Supplementary
Video).

### 2.2.2 Sea ice concentration time series

We complement the SAR data snapshots with a more complete, but lower spatial resolution time series of daily sea ice concentration provided by the University of Bremen's sea ice data center (Spreen et al., 2008). The sea ice concentrations are derived from the microwave radiometer data of the Advanced Microwave Scanning Radiometer 2 instruments onboard the
Japan Aerospace Exploration Agency Global Change Observation Mission-Water satellite using the ARTIST Sea Ice algorithm (Spreen et al., 2008). This algorithm primarily uses the difference between the brightness temperatures for the V and H polarizations at 89 GHz for the calculations. We used the Antarctic daily product (asi_daygrid_swath) with no land





mask applied and processed to 3.125 km grid spacing (Antarctic3125NoLandMask). We apply the Norwegian Polar Institute Quantarctica 3 Basemap (ADD_Coastline_high_res_polygon_Sliced) land and ice shelf masks around PIB to retrieve only

concentrations over open ocean and calculate the daily mean sea ice concentration (%) across the PIB sea ice sampling box (102° - 106° W, 74.5° - 75.0° S; dashed red box in Fig. 10) from January 2020 to January 2023.

### 2.3 Wavelet analysis

Our dataset exhibits variability across multiple time scales, with certain signals emerging or fading throughout the duration of the record. We employed cross wavelet transforms on the hydrographic records to uncover any systematic temporal

patterns in their variability. This was carried out using the MATLAB package developed by Grinsted et al. (2004). Unlike traditional harmonic analysis integrating signals over time, wavelet analysis has the advantage of identifying changes in power over time for a specific period. The cross-wavelet transform shows regions in time-frequency space where two time series share high common power, indicating periods of statistically significant covariance. Thus we resolve intermittent signals across sub-daily periods as well as longer-period ones, spanning up to several months. Furthermore, the cross wavelet

transforms explore potential phase discrepancies among $\Theta$ and $S_A$ time series, which indicate whether one leads or lags the other. These were visualized with quivers where the arrow direction indicates if one time series leads the other at that specific period or if they occur harmonically in phase. The statistical significance of the identified periodicities in covariance was determined using standard Monte-Carlo methods against red noise background (see Grinsted et al., 2004). Before computing cross-wavelet transforms, we linearly interpolated the data onto evenly-spaced temporal resolution increments of

10 minutes, applied a Chebyshev low-pass filter to eliminate any outliers and detrended the time series. The cut-off period of the Chebyshev filter consequently sets the minimum signal that can be resolved with the wavelet transform. Given that the Amundsen Sea exhibits a diurnal tidal regime, we applied a cut-off period of 0.125 days (or 3 hours) for the Chebyshev filtering to resolve the tidal variability in our datasets.

### 3 Results

### 3.1 Ocean variability beneath TEIS

Hydrographic properties show variability across a wide range of timescales (Fig. 2). $\Theta$ and $S_A$ increase with depth, with mean $\Theta$ of -0.88 ± 0.24 °C at 316 m, 0.34 ± 0.09 °C at 521 m, and 1.04 ± 0.04 °C and 1.05 ± 0.03 °C near the seafloor at depths of 745 m and 784 m, respectively (Fig. 2). We observe a warming trend at all depths relative to these mean values until July 2021. After this, warming stalled at depth, while mid-depth and shallow layers cooled until January 2022.

Thereafter, warming resumed at mid-depth and both deeper layers, continuing through to January 2023.

From January 2020 to July 2021, the shallow MicroCAT recorded a 1°C increase in $\Theta$ at a rate of 0.4 °C yr$^{-1}$, followed by a 1 °C decrease at an accelerated rate of -1.8 °C yr$^{-1}$ until the instrument ceased operation in January 2022 (Fig. 2a). After





July 2021, fluctuations in $S_A$ became more pronounced, consistently exceeding the overall mean of 34.23 g kg$^{-1}$ and exhibiting a declining trend from July 2021 to January 2022. The Pearson correlation coefficient between $\Theta$ and $S_A$ at the shallow instrument was 0.4 before July 2021, increasing to 0.7 afterwards.

The mid-depth MicroCAT recorded a 0.1 °C increase in $\Theta$ over the entire record, although in a stepped fashion (Fig. 2b). The warming trend was 0.2 °C yr$^{-1}$ until July 2021, steepening notably between March and July 2021, when $\Theta$ and $S_A$ increased in tandem. This was followed by a gradual decline beyond their initial values at a rate of -0.5 °C yr$^{-1}$ until January 2022, after which warming resumed at 0.2 °C yr$^{-1}$ until January 2023.

Both deep MicroCATs recorded a 0.1 °C warming from April 2020 to January 2023, accompanied by a 0.02 g kg$^{-1}$ increase in $S_A$ (Fig. 2c,d). $\Theta$ and $S_A$ fluctuations were generally synchronous at both deep MicroCATs. Near the seabed at Cavity Camp, warming occurred at a rate of 0.04 °C yr$^{-1}$ until July 2021, then plateaued until January 2022, after which it resumed warming at a rate of 0.01 °C yr$^{-1}$ until January 2023. At Channel Camp, the warming trend near the seabed was also 0.04 °C yr$^{-1}$ until July 2021, then plateaued before increasing to 0.02 °C yr$^{-1}$ after January 2022. This suggests that between January 2022 and January 2023, the warming trend re-emerged in both mid-depth and deep layers.

Between January 2020 and January 2022, both shallow (315 m) and deep (782 m) sensors at Channel Camp sank at rates of 2.21 m yr$^{-1}$ and 2.17 m yr$^{-1}$, respectively (Appendix C). A background sinking rate of approximately 1.86 m yr$^{-1}$ is expected from compaction of firn underneath the AMIGOS-3. The shallow MicroCAT stopped recording on January 11, 2022, at 319 m depth, while the deep sensor continued operating until January 2, 2023, reaching 788 m (Fig. C1). Notably, the sinking rate of the deep sensor decreased to 1.62 m yr$^{-1}$ during 2022, indicating a possible reduction in the density of the overlying water column and a concurrent decline in firn compaction. At Cavity Camp, the mid-depth MicroCAT was initially deployed at 520 m and the deep MicroCAT at 744 m. Both began recording on January 2, 2020, and continued until December 26, 2022, reaching depths of 523 m and 747 m, respectively (Fig. C2). The consistent sinking trends observed at each site, along with the strong agreement between pressure records from sensors at the same site, rule out the possibility that the mooring cables became anchored to the seafloor.







**Figure 2:** Time series of anomalies in conservative temperature (Θ) and absolute salinity (S$_A$) at (a) 316 m at Channel Camp, (b) 521 m at Cavity Camp, (c) 745 m at Cavity Camp and (d) 784 m at Channel Camp. Mean values of Θ and S$_A$ are indicated in the respective legends. Gray bars indicate periods when the measured current speeds were elevated. No additional Aquadopp current meter data are available after March 2021.

Wavelet analysis reveals that the MicroCAT Θ and S$_A$ co-vary at all depths and across all periods (rightward arrows in Fig. 3), supporting the use of Θ anomalies as proxies for salinity for the DTS time series. At shallow depths, statistically significant covariance with periods longer than 8 days is minimal before July 2021 (Fig. 3a). However, from July 2021 until the end of the shallow record in January 2022, covariance appears at periods of up to 24 days. At mid-depth, similar covariance with periods up to 24 days emerges in April 2020, with occasional occurrences of significant covariance lasting more than a week observed in September 2020 (Fig. 3b). Following this, multi-day covariance shifts primarily to sub-daily



covariance for most of the remaining record. At greater depths, statistically significant covariance with periods lasting several months is observed, especially at Cavity Camp (Fig. 3c). This longer-term covariance diminishes after July 2021, gradually shifting toward shorter periods of around one week by January 2023.

Notably, the long-term covariance at depth is overlaid by significant diurnal and semi-diurnal fluctuations, which are also more prominent at Cavity Camp than Channel Camp (Fig. 3c,d). The shorter-term variability is closely tied to the prevailing tidal regime, which is predominantly diurnal with some semi-diurnal components. Significant tidal periods exhibit enhanced power with a fortnightly modulation, indicating influence from the 14-day spring-neap tidal cycle. We observe, however, only little covariance at tidal periods in most of the shallow record and throughout the mid-depth record, whereas tidal covariance is evident at both deep sites (Fig. 3c,d).

Superimposed on the long-term variability, we observe several distinct events, characterized by rapid $\Theta$ and $S_A$ excursions over several weeks, notably in April and July 2020, as well as in February and April 2021. During these events, concurrent decreases in $\Theta$ and $S_A$ of more than 0.05 °C and 0.03 g kg$^{-1}$, respectively, were recorded at the deep sites. The mid-depth and shallow instruments simultaneously displayed opposite signals, with rising $\Theta$ and $S_A$ anomalies of more than 0.3 °C and 0.04 g kg$^{-1}$, and 0.2 °C and 0.03 g kg$^{-1}$, respectively. Simultaneous current velocity measurements revealed accelerated current speeds at all depths during those events (grey-shaded time spans in Fig. 2).





**Figure 3:** Cross-wavelet transform between temperature and salinity time series at (a) shallow, (b) mid-depth, and deep sensors (c) at Cavity Camp and (d) Channel Camp. Warm colours show high power at the corresponding period. Black contours depict statistical significance. Arrows show the phase relationship between $\Theta$ and $S_A$ covariance (all pointing right means both occur in phase during significant periods). Greyed out is the cone of influence where edge effects might obscure the cross-wavelet transform.

To identify the water sources advecting through our sensors at shallow, mid-depth, and deep layers, we compare our MicroCAT CTD data recorded from January 2020 to January 2023 with two AUV datasets collected at T2 and T3 in February and March 2019 as well as a set of ship-based CTD measurements from PIB collected during the same cruise. The



Θ-S$_A$ diagrams reveal that PIB-sourced water is generally the warmest throughout the water column, followed by T3 and T2 (Fig. 4). At depth, our measurements from both sites align most closely with those from PIB (Fig. 4d–f). The observed events at depth are characterized by cold and fresh water types (blue arrows in Fig. 4d) that are not typically present in the

established deep-water masses. Notably, a distinct hook in our deep-layer data, observed at both Cavity Camp and Channel Camp, follows a constant density of 1027.8 kg m$^{-3}$ (red arrows in Fig. 4e). This characteristic, also present in the AUV data from T3, was previously traced to PIB by Wåhlin et al. (2021) and results from isopycnal mixing between PIB and Thwaites Trough water, indicating the far western extent of PIB influence. The slope of this hook is also represented in our data, even more prominently than in the T3 dataset, though with a slight offset in S$_A$ (Fig. 4e). Additionally, none of our measurements

overlap with the coldest water masses observed at T2 in Θ-S$_A$ space, reinforcing the hypothesis of Wåhlin et al. (2021) that cooled, meltwater-enriched water exits the subshelf cavity via T2.

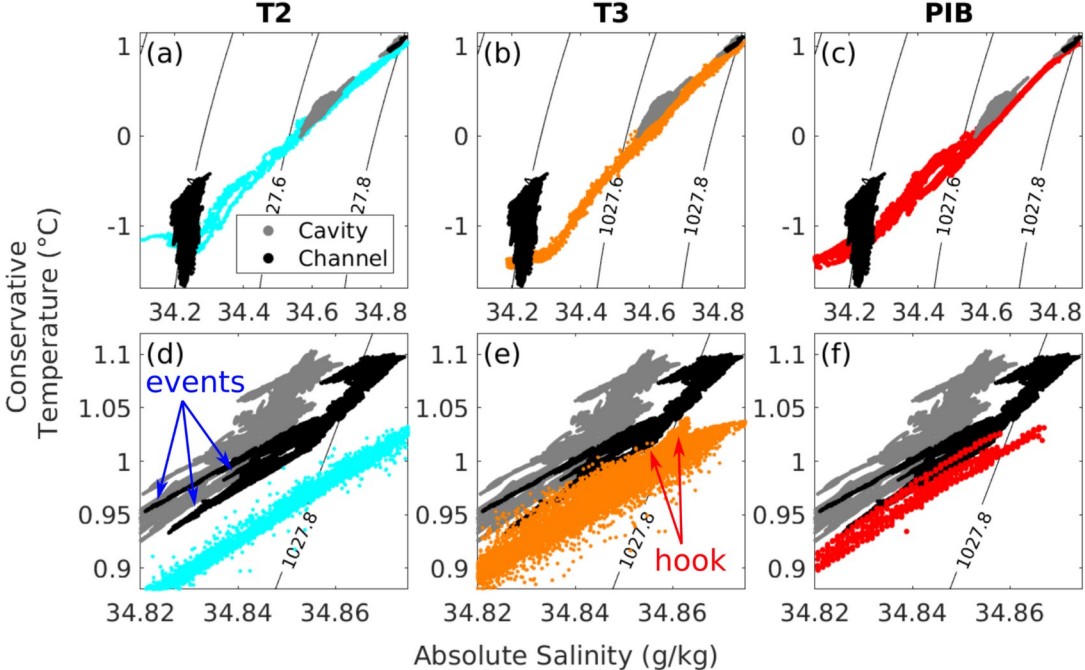

**Figure 4:** Θ-S$_A$ diagrams from MicroCATs at Cavity and Channel Camp (grey and black), compared to AUV measurements at (a) T2

(blue), (b)T3 (orange), and (c) ship-based CTD (red) in PIB. See Figure 1 for a map of these locations. (d-f) Close-up views of the mCDW layer at depth, with isopycnals representing lines of potential density (kg m$^{-3}$). Labels refer to features discussed in the text.

The Gade line represents the mixing between glacial meltwater and mCDW, where small salinity changes correspond to significant temperature variations due to heat and salt exchange during ice melting (Gade, 1979). The mCDW-Winter Water

(WW) mixing line, on the other hand, reflects the dilution of WW with mCDW. WW is characterized by a subsurface temperature minimum and represents the remnant of the winter surface mixed layer, which becomes capped in summer by



fresher and warmer water due to sea ice melt and air-sea heat fluxes. At the shallow MicroCAT, water masses gradually shift toward the Gade line from January 2020 to January 2021 and closely follow it until July 2021 (Fig. 5a). Thereafter, they align with the 1027.42 g kg$^{-1}$ isopycnal, indicating reduced glacial meltwater influence due to increased WW advection into

the TEIS subshelf cavity. At mid-depth, data cluster along a linear trend between the Gade and WW mixing lines, suggesting a stable water mass structure with a gradual warming and freshening trend (Fig. 5b). At depth, waters follow a narrow mixing path between these two lines, with long-term warming and salinification. The highlighted events, where $\Theta$ and $S_A$ drop for several weeks, align with the Gade line (blue arrows in Fig. 5c), while the long-term evolution of the densest waters follows an extension of the WW mixing line. This characteristic "hook" shape (red arrow in Fig. 5c), previously identified by

Wåhlin et al. (2021), is indicative of mCDW originating from T3 (Fig. 4e).

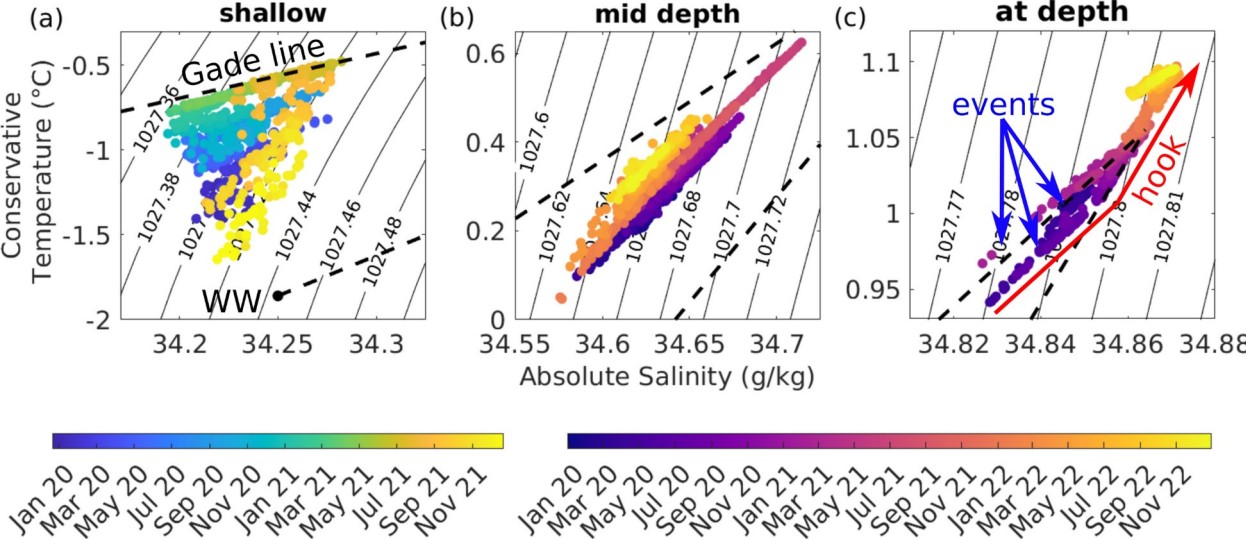

**Figure 5:** $\Theta$-$S_A$ diagrams from MicroCATs at Cavity Camp and Channel Camp, showing changes in water mass composition and mixing over time. (a) The shallow record (316 m) covers only the period from January 2020 to January 2022, while (b) the mid-depth (521 m) and
(c) deep records (784 m) extend from January 2020 to January 2023. In all panels, the upper dashed line represents the Gade line, indicating water mass modification through ice-shelf melting, while the lower dashed line is the WW mixing line, showing the influence of cold surface water mixing. Solid black lines represent isopycnals of potential density (kg m$^{-3}$). Labels refer to features discussed in the text.

### 3.2 Wind and ocean current dynamics and their influence on hydrographic variability

Winds sweeping across the ice-shelf surface predominantly originate from the ESE (Fig. 6a). The average wind speed at Channel Camp was 10 m s$^{-1}$, with occasional spikes surpassing 60 m s$^{-1}$ during winter or early spring. In-situ and ERA5 air temperature and wind speed showed strong agreement, whereas wind direction data agreed to a much lesser extent (Appendix A).





The ocean currents beneath TEIS are usually slow. At the shallow Aquadopp, the mean current direction was
predominantly toward the SSW (211° ± 71°), with an average speed of 2.2 ± 1.8 cm s$^{-1}$. The mid-depth Aquadopp recorded a
mean current flowing toward the SSW (221° ± 58°) at an average speed of 3.7 ± 2.2 cm s$^{-1}$. The deep Aquadopp at Cavity
Camp (745 m) exhibited a mean current toward the SSW (227° ± 64°) at an average speed of 0.9 ± 0.7 cm s$^{-1}$, while the deep
Aquadopp at Channel Camp (784 m) showed a mean current directed toward the N (8° ± 136°; note the higher current
direction variability than the other sensors) at an average speed of 0.8 ± 0.8 cm s$^{-1}$.

During the April 2020 event, currents at the shallow Aquadopp intensified, reaching speeds exceeding 7 cm s$^{-1}$ and
flowing toward NNW (Fig. 6b). At mid-depth, currents accelerated to a similar magnitude but flowed toward the SW (Fig.
6c). In the deep layer, currents also flowed toward SW, with a maximum recorded speed of 4.6 cm s$^{-1}$ on April 18, 2020 (Fig.
6d,e). Another event occurred in July 2020, when the shallow Aquadopp at Channel Camp recorded an accelerated current of
9 cm s$^{-1}$, now flowing toward the SSW. However, this event was not clearly observed at the deep Aquadopp at Channel
Camp, and data gaps from both Aquadopps at Cavity Camp prevent further investigation. The most widespread event
occurred in February 2021, when all four Aquadopps recorded elevated current speeds. The shallow Aquadopp measured
persistent currents of ~9 cm s$^{-1}$ toward the SSW, while the mid-depth Aquadopp recorded even higher speeds of ~11 cm s$^{-1}$
directed SE. At Cavity Camp, the deep Aquadopp peaked at 4 cm s$^{-1}$ toward the SW on February 7, 2021, whereas the deep
Aquadopp at Channel Camp exhibited a contrasting current direction of 5 cm s$^{-1}$ toward the NW. The Aquadopps ceased
operation before the fourth temperature and salinity drop in April 2021, preventing the determination of dominant current
directions for this event. At all depths, prolonged temperature and salinity anomalies, likely accompanied by enhanced
current speeds, ended after May 2021 and were replaced by increased shorter-period covariance (0.5 to 16 days).

To determine if the changes in hydrography are driven by ocean currents, we performed the cross wavelet transform
between water density and current speed. For the shallow and mid-depth sensors increasing current speeds co-vary with
increasing density, while at depth increasing current speed co-varies with decreasing density. We identify significant long-
period covariance between one to four weeks in April 2020 and in July 2020. All sensors show covariance from sub-daily to
multi-weekly time periods in February 2021 (Fig. 6g-j). We also find significant multi-week covariance between ERA5 wind
speed and density variations at the shallow ocean sensor in April and July 2020 (Fig. 6f).





**Figure 6:** Feather plots of average daily (a) in-situ wind and (b-e) current speed and direction from January 2020 to March 2021. The line orientation represents wind and current direction (with the top of the graph indicating North or 360°), while line length corresponds to speed. Wind direction follows the meteorological convention, indicating the direction from which the wind originates, whereas currents are shown flowing toward their respective directions. The grey shaded areas denote periods of elevated current speeds as discussed in the text. (f) Cross-wavelet transform between shallow density and ERA5 windspeed covariance. (g-j) Cross-wavelet transforms between density and current speed time series for each depth.





The DTS temperature profiles at Channel Camp during the four highlighted events reveal a consistent pattern of temperature changes within the water column (Fig. 7). Throughout all events, water masses between 400 and 600 m depth exhibit warming, with the most pronounced temperature increase occurring around 450 to 500 m depth. Conversely, the deeper

water between 600 and 800 m experiences cooling, which is strongest at 700 m depth. Additionally, a near-isothermal layer forms between 300 and 400 m, suggesting vertical mixing in this depth range. The temperature profiles show a progressive shift in thermal structure, with warming and cooling trends developing simultaneously in distinct layers. Notably, the 600 m depth emerges as a clear transition point, marking the boundary between the warming upper layers and the cooling deeper waters.


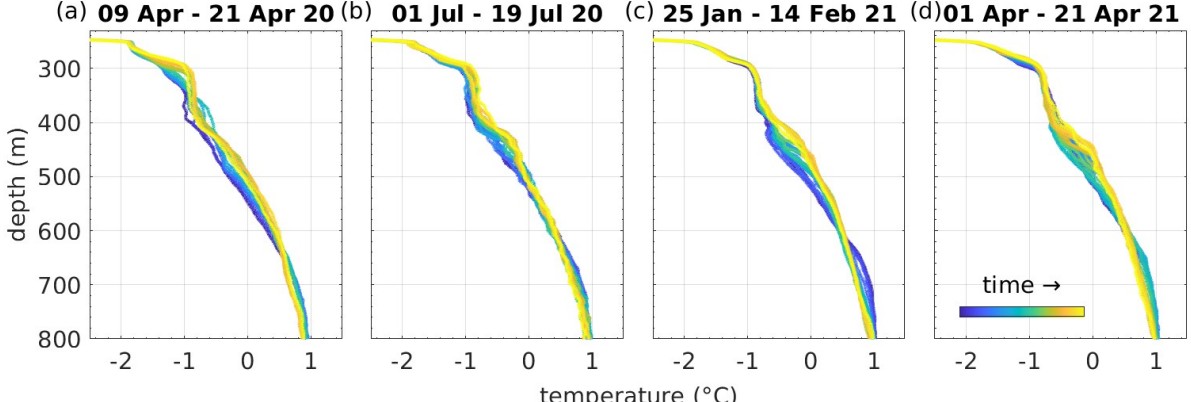

**Figure 7:** Daily mean DTS temperature from Channel Camp profiles for the specified events. The plots reveal a warming trend in the upper two-thirds of the water column, accompanied by cooling in the lower third. The profiles are colour-coded, transitioning from cool to warm colours, to represent the progression of time.


To estimate the length scale of the advecting features, we combine DTS profiles with Aquadopp current speed measurements (Fig. 8). Specifically, we combined the mid-depth Aquadopp (521 m) at Cavity Camp with the bottom Aquadopp (784 m) at Channel Camp to calculate daily mean current speeds, which we then assumed to be representative throughout the water column. The event in April 2020 reveals a feature with an approximate length of 30 km, while the July 2020 event shows a

feature of about 20 km in length, although a data gap during the austral winter prevented capturing the full scale of this feature. The feature observed in February 2021 is the largest and most clearly defined in our dataset, with a length scale of around 100 km. Malfunctioning Aquadopps in March 2021 prevented the assessment of the feature in April 2021. All captured features show a ~400 m vertical extent.

Isopycnals, estimated from combining the DTS profiles with salinity from CTD profiling on January 12, 2020, show

that the warming observed between 400 m to ~600 m depth leads to thermal expansion of the water column, while the cooling observed between 600 to 800 m depth pushes isopycnals down, but to a much lesser extent (Fig. 8). This is not



surprising because at depth, changes in density are driven primarily by changes in salinity, which do not show a large vertical gradient (Appendix B).

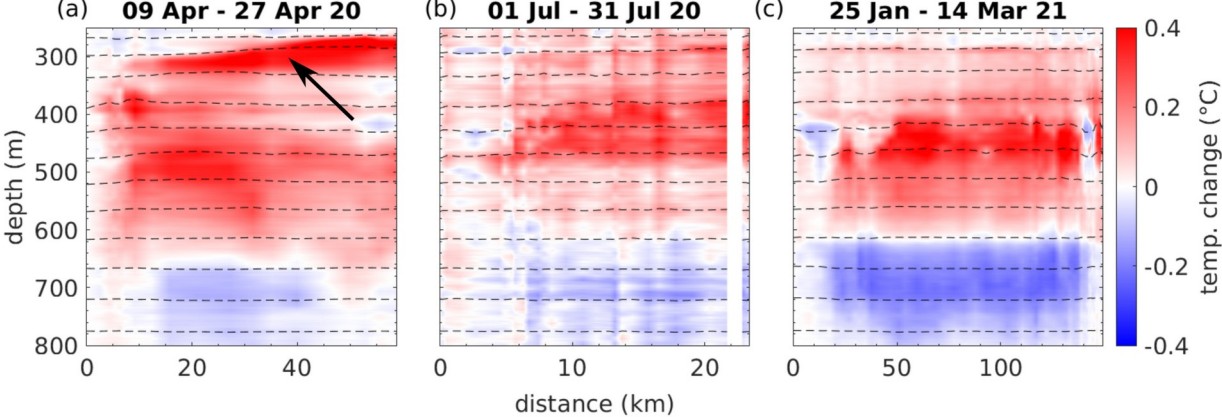


**Figure 8:** Temperature anomalies over time for three distinct periods. Each panel shows the deviation from the first profile in the respective period. The colour scale represents the magnitude of temperature change, with negative values indicating cooler temperatures at depth and positive values indicating warmer temperatures above ~600 m depth. The x-axis reflects the distance travelled by features advecting through the water column, based on Aquadopp current speed measurements, available during the first three events. Dashed black
lines show isopycnals. The black arrow in panel (a) shows warming in the shallowest layer discussed in the text.

### 3.3 Thermodynamics in the water column

The DTS data provide a continuous vertical record of ocean temperatures. Both mooring sites feature an approximately 100 m thick layer of mCDW near the bottom that exhibits temperatures exceeding 1.1 °C. This bottommost layer is not only
warming with time (Fig. 2c/d), but also thickening by about 50 m in its vertical extent throughout the record (Fig. 9). Situated above this warmest layer, a 200 m thick zone demonstrates a sharp thermocline between 500 and 700 m depth, with temperatures generally above 0 °C. Further up the water column lies another 200 m thick layer (300 to 500 m deep), characterized by temperatures between -1 and 0 °C. At the Channel Camp site (Fig. 9a-c), within a narrow band spanning the next 40 m, a thin layer approaches -1.5 °C, nearing the in-situ freezing point at approximately -2 °C. This cold layer thins
between January 2020 and July 2021 at this site. In the immediate vicinity of the ice-shelf base, a 2-3 m thin layer at the pressure melting point (-2 °C) is observed. This insulating layer, which has also been documented in proximity to the ice-shelf grounding zone at greater depth, effectively suppresses basal melting through strong stratification (Davis et al., 2023). The ice base with a draft of 260 m lies above the depth of the mCDW which is greater than 600 m. Even without the insulating layer, the thermal driving is low and insufficient to sustain significant basal melt rates.
415       The DTS record at the Channel Camp site suffers a substantial data gap from August 2021 to January 2023 (Fig. 9a), but reveals a significant cooling trend of more than 1.2 °C in the upper half of the water column across that gap (Fig. 9c).





This cooling phenomenon in the 250 m directly beneath the floating ice contrasts with the continuous DTS record prior to the data gap, suggesting considerable changes in the subshelf hydrographic properties. Notably, the 40-m-thick cold layer, nearing the in-situ freezing point that is observed in the August 2021 profile, expanded to a 150-m-thick layer (250-400 m depth) in the January 2023 profile (Fig. 9c). Between 400 and 500 m depth, a sharp temperature gradient of 0.013 °C m$^{-1}$ is observed. However, the lower half of the water column exhibits temperatures similar to those observed in August 2021, suggesting that the water masses in the lower half of the water column persisted, while the upper half experienced a considerable change in hydrographic properties. This decrease in temperature corresponds to a change in mean water column density from 1029.3 to 1029.1 kg m$^{-3}$, assuming no change in salinity between 250 to 500 m depth (Appendix B). The lightening of the upper half of the water column aligns with the pressure changes observed at the deep CTD, which sank at a rate of 2.17 m yr$^{-1}$ between 2020 and 2022, and at a reduced rate of 1.62 m yr$^{-1}$ in 2022.

The DTS record at Cavity Camp is similar to the observations from Channel Camp but provides additional data from August to the end of October 2021, after which no further DTS measurements were taken at this site. Notably, the Cavity Camp DTS recorded the onset of the cooling of the upper water column (Fig. 9d). By analyzing the last 100 DTS profiles dating back to June 2021, we determined that the cooling occurred rapidly in late July 2021, reaching a depth of approximately 450 m before the DTS record ended by early October 2021 (Fig. 9e,f).





**Figure 9:** Daily-binned temperature records from DTS at (a) Channel Camp and (d) Cavity Camp. (c) Last temperature profile before the August 2021 - January 2023 gap and the first measurement in January 2023, highlighting cooling in the upper water column. Dotted, dashed, and solid black lines indicate the depths of shallow, mid-depth, and deep ocean sensors. (b) and (e) Waterfall diagram of the last 100 DTS profiles at Channel Camp and Cavity Camp, showing abrupt cooling between 300 and 400 m depth. The temperature range of each line is presented in (f), with an example of the last DTS profile from October 2021 (red). Note that there is a period with no data in August and September 2021 at Cavity Camp. The DTS profiles shown in the waterfall plots were smoothed for visualization with a running mean of 40 sample points (corresponding to ~10 m along the cable).




### 3.4 Sea ice conditions in PIB: formation and breakup of fast ice


We examine the multi-year evolution of sea ice coverage in PIB to identify the potential drivers of variability in hydrographic properties beneath TEIS. At the start of our observational period in the austral summer of 2019/20, PIB was largely free of sea ice, with open water extending from TEIS to the ice front of Pine Island Glacier (Suppl. Video). As surface air temperatures dropped below -10 °C through March 2020 and winds remained generally calm (Fig. A1), thin first-

year sea ice began to form (Fig. 10a/b). By late March and into April 2020, a major sea ice breakout event occurred, driven by strong easterly winds exceeding 20 m s$^{-1}$. These winds fractured the newly formed ice and redistributed it, revealing an active PIB gyre in satellite SAR imagery, marked by the cyclonic movement of sea ice (Fig. 10c). By mid-April winds calmed to around 5 m s$^{-1}$ and air temperatures stayed below -10 °C (Appendix A), promoting sea ice formation by latent heat loss and leading to near-complete sea ice coverage in PIB (Fig. 10d). This coverage persisted through the following two

austral summers (2020/21 and 2021/22).

During the April 2020 sea ice breakout, we observed the first event of opposing density anomalies between the shallow/mid-depth and deep sensors (Fig. 10f/g). Similar anomalies occurred in July 2020, as well as in February and April 2021, when thin first-year sea ice is moving around PIB. However, these events disappeared after May 2021, when the now second-year sea ice more firmly fastened across PIB. The fast ice cover remained until January 2022 after which the fast ice

front gradually retreated, eventually breaking up in October 2022 and leading to open-water conditions in PIB once again by February 2023 (Fig. 10e).



**Figure 10:** Co-evolution of PIB sea ice and Thwaites sub-ice shelf ocean densities. Panels (a-d) present Sentinel-1A SAR images depicting a first-year, sea ice breakout occurring between mid-March and late-April 2020. Panels (e-h) show the retreat of the multi-year, fast-ice edge to the grounding line of Pine Island Glacier. The dashed red rectangle shows the sea ice concentration sampling box. The black line indicates the position of the ice-shelf front and grounding line (Bindschadler et al., 2011). Red and blue dots denote Channel Camp and Cavity Camp locations on TEIS. Panel (i) shows sea ice concentration time series in PIB. Panels (j) and (k) display time-series data of ocean water density anomalies at these sites across various depths. Grey dashed lines indicate the times of SAR image capture shown in panels (a-h).





## 4 Discussion

Our results support the narrative of Zheng et al. (2022) that variability in subshelf oceanography is influenced by sea ice conditions in PIB. The novelty of our study lies in the finding that different sea ice types lead to characteristic signatures in the subshelf water column. Mobile unconfined sea ice generates surface stress on the ocean, driving circulation similar to wind forcing on open ocean water (Fig. 11a/b). Strong winds in PIB lifts mid-depth isopycnals and facilitates the formation of gyre-scale features (tens of kilometers) which are subsequently advected beneath TEIS, altering the thermal structure between 400 m and 800 m depth over several weeks. In contrast, when PIB is covered by persistent, near-stationary, or landfast multi-year sea ice, wind stress transfer into the ocean is inhibited (Fig. 11c), preventing the formation and advection of these mid-depth features. An extended duration of fast ice coverage leads to overall warmer conditions beneath TEIS (Dotto et al., 2022) and the accumulation of meltwater in the upper ocean layers, driven by sub-ice-shelf melting and buoyant meltwater from the deep grounding lines Pine Island and Thwaites Glaciers (Fig. 11d). As the ice edge retreats, colder WW is advected beneath the ice shelf in the upper layers, while variability in mCDW at depth occurs primarily on tidal timescales (Fig. 11e), contrasting with the longer variability observed under sea ice-covered conditions. We hypothesize that after the fast ice breakout in January 2023, when data collection ended, the mid-depth features reappear as sea ice and ocean conditions continue to evolve.

We propose that these events are driven by heaving and sinking around an expanding layer at 600 m depth, which marks the top of the mCDW layer. During the events, water masses extend both upward and downward (Fig. 7), suggesting the influence of gyre-scale features moving through the water column and driving the transient vertical expansion of water masses (Fig. 8). This interpretation is supported by the DTS profiles, which reveal periodic expansions of water masses centered around 600 m depth (Fig. 9a/d). Additionally, during these events, the hydrographic properties shift back and forth along a defined trajectory, indicating that no mixing of water masses occurs. Instead, the variability is driven by vertical isopycnal displacement of the same water mass (Fig. 5c).





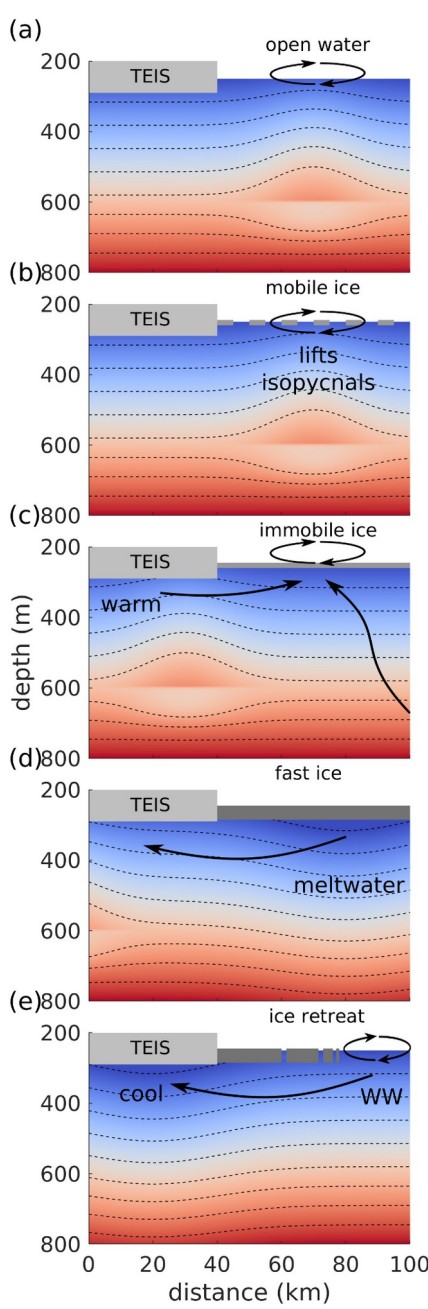

**Figure 11:** Schematic representation of the interactions between sea ice dynamics and hydrographic variability.



## 4.1 Gyre-scale features formed during mobile, first-year sea ice breakouts

In April 2020, the shallowest layers experienced significant warming following the passage of a gyre-scale feature (Fig. 8a). At 316 m depth, where the shallow CTD is located, we observe an accelerated NNW-directed outflow (Fig. 6b). During this time, PIB is covered by mobile, first-year sea ice (Fig. 10a-d), and southerly winds blow across the ice-shelf surface (Fig. 6a). Density at shallow levels increases steadily throughout the month by approximately 0.04 g kg$^{-1}$ (Fig. 10f). Cross-wavelet analysis reveals significant covariance between wind speed and density fluctuations at shallow depths (Fig. 6f), as well as

between current speed and density (Fig. 6g). This suggests that winds drive surface waters away from the ice-shelf front toward open water. Sea ice formation through latent heat loss to the atmosphere then leads to brine rejection and explains the increase in near surface density. During the subsequent events in July 2020 and February 2021, the warming signal at shallow levels associated with the advecting features between 400 and 800 m depth is not observed (Fig. 8b/c). We therefore interpret the warming in the uppermost layer during the first event as a wind-driven, localized anomaly, likely facilitated by

the open water surface to the north of Thwaites Pinning Point (Fig. 1a).

In July 2020, we observed a subsurface feature without any associated warming in the shallowest layers (Fig. 8b). Unlike the April 2020 event, the currents at shallow depth were directed toward the SSW, indicating that the observed feature was advected from the NNE beneath TEIS (Fig. 6b). ERA5 shows significant covariance between wind speed and density fluctuations at shallow depths on timescales exceeding one month (Fig. 6f), as does the relationship between current

speed and density (Fig. 6g). Unfortunately, this period is only covered by the Aquadopps at Channel Camp (Fig. 6b/e), which confirm significant covariance between current speed and density fluctuations at shallow levels but not at depth (Fig. 6g/j). The DTS record at Channel Camp captured most of this event, showing warming between 400 and 600 m depth and cooling between 600 and 800 m in early July (Fig. 7b). However, the DTS record ends in early August 2020, before the event concluded (Fig. 8b). We interpret this event as being driven by wind stress toward the NNE, where open water and

mobile, first-year sea ice were still present to transmit the prolonged wind forcing into the ocean, while PIB remained covered by mobile, first-year sea ice (Fig. 10e).

In February 2021, we captured the clearest event occurring between 400 and 800 m depth (Fig. 8c). Similar to the July 2020 event, shallow currents were directed toward the SSW (Fig. 6b). However, unlike July 2020, current speed variability at 316 m depth did not significantly co-vary with ERA5 wind speeds (Fig. 6f) or with density variability at this depth. This

suggests that the near-isothermal layer, observed between 300 and 400 m depth (Fig. 7c), likely formed due to turbulent mixing, independent of the deeper event. At mid-depth, and within the warming part of the water column (400–600 m), currents flowed toward the SSE, with speed variability driving density fluctuations on timescales exceeding two weeks (Fig. 6h). At greater depths, within the cooling part of the feature (600–800 m), currents shifted from SSE at mid-depth to SSW at depth (Fig. 6d). Current variability at Cavity Camp influenced density fluctuations on timescales of up to a month (Fig. 6i),

with an even clearer signal at Channel Camp (Fig. 6j). During this period, PIB remained covered by first-year sea ice (Fig. 10e), while open water areas with mobile sea ice were present north of Thwaites Pinning Point. We therefore suggest that the



captured feature in February 2021 also originated from the open water surface to the north of Thwaites Pinning Point before advecting beneath TEIS.

## 4.2 Conditions during immobile, multi-year fast ice cover

After May 2021, no further events were observed at mid-depth. During this period, the second-year sea ice in the PIB reached its maximum extent, becoming fastened between the ice edge of TEIS to the west and Antarctica's coastline to the east. This fast-ice platform stretched from Thwaites Pinning Point to the grounding line of Pine Island Glacier. The extensive, immobile fast ice effectively isolated the ocean from atmospheric wind stress. Hydrographic data reveal an increasing meltwater content at both shallow and mid-depth levels until July 2021 (Fig. 5a/b). This observation aligns with 530 the findings of Dotto et al. (2022), who suggest that prolonged fast ice coverage in PIB facilitates the accumulation of meltwater beneath the ice cover, extending beyond TEIS. This meltwater likely originates from a combination of subshelf melting beneath TEIS and melting along the deep grounding lines of Thwaites and Pine Island Glacier. The resulting meltwater-enriched plumes rise through the water column due to their relative buoyancy, reaching shallower layers. Unfortunately, all Aquadopp current meters malfunctioned during this period, preventing a determination of the source 535 region for these water masses.

## 4.3 Fast-ice breakout and increased WW advection

The retreat of the fast ice edge began at the end of the austral summer in January 2022 (Fig. 10e), when a significant portion of multi-year fast ice in northeastern PIB broke up, exposing open water (Fig. 10e/f). During the following winter, surface cooling from atmospheric conditions likely allowed WW to recharge in this open-water region, contributing to the observed 540 cooling in the upper half of the water column within the TEIS cavity (Fig. 9c). However, whether WW originated specifically from this newly exposed area or was supplied by enhanced advection of a colder WW variety remains uncertain, as both processes could explain the observed cooling in our DTS record and WW properties change both from year to year, and spatially.

Evidence supporting WW advection, rather than cooling driven by meltwater-enriched water masses, comes from the 545 shallow MicroCAT, which indicates a concurrent decrease in the mCDW-derived meltwater content toward the WW mixing line in late 2021 (Fig. 5a). Another possible explanation for the cooling is increased subglacial outflow, but grounding-line discharge is typically associated with lower salinity and minimal change in potential temperature (Davis et al., 2023). Given these factors, we conclude that enhanced WW advection is the most likely cause of the observed cooling.

## 4.4 Potential formation mechanisms of the observed events

Different types of sea ice play a significant role in shaping the oceanographic variability beneath TEIS, supporting the ideas presented by Zheng et al. (2022) and Dotto et al. (2022). The main distinction from Dotto et al. (2022) is the use of a longer





oceanographic record that captures changes in hydrographic properties as sea ice cover in PIB evolves, along with a more extensive use of the DTS dataset to examine the vertical extent and timing of changes within the subshelf cavity. While the authors suggested that a cyclonic PIB gyre lifts isopycnals in PIB, causing them to sink beneath TEIS and resulting in colder

conditions, our study reveals a delayed, contrasting response at depth. We observe warming between 400 and 600 m depth and cooling between 600 and 800 m following an active, cyclonic PIB gyre. The PIB gyre spans approximately 50 km and transports around 1.5 Sv of water, reaching depths of about 700 m (Thurnherr et al., 2014). Our observed events are centered around 600 m depth, which may explain the upward displacement of isopycnals above this level. However, the mechanism responsible for the opposing effect at greater depths remains an open question and requires further investigation within a

well-defined numerical framework. This interface layer, located at 600 m depth, also marks the top of the underlying mCDW layer.

Recent numerical simulations of the Amundsen Sea suggest that the ice-shelf cavities beneath Thwaites and Pine Island Ice Shelves are favorable environments for submesoscale eddies (O(0.1–10 km), O(1 day); Shresta et al., 2024). These eddies transport heat vertically toward the ice shelf base, potentially enhancing basal melting in a positive feedback loop

(Shresta et al., 2024). However, identifying their formation mechanisms remains challenging due to the lack of direct observations within the ice-shelf cavity. We anticipate that our dataset will help constrain these mechanisms. The features we observe, however, exhibit larger horizontal and temporal scales (O(10–100 km), O(1 month)) and a greater vertical extent (O(100 m)) compared to the O(10 m) submesoscale eddies simulated by Shresta et al. (2024). Additionally, while their modelled eddies formed behind bathymetric sills at depth, lifting mCDW upward, our observed features display an opposing

signal, centered around 600 m depth, temporarily pushing mCDW downward.

Fluctuations in thermocline depth, where temperatures rapidly increase from 0 °C to +1 °C, separating the cold WW above from the warm mCDW below, have been linked to wind stress variations over the open ocean in PIB. These fluctuations have been associated with changes in basal melt rates beneath Pine Island Ice Shelf on a similar timescale to the features we observe (O(1) month, Davis et al., 2018). While wind stress primarily drives isopycnal displacement within the

thermocline, where vertical density gradients are strongest, this mechanism produces a uniform response throughout the water column and does not explain the opposing trends we observe, which instead manifest as periodic thickening centered around 600 m depth.

Mooring observations near the front of Getz Ice Shelf have shown that WW deepening beyond 550 m is associated with strong easterly winds and reduced sea ice cover, originating about 100 km from the mooring site. This process

generates intra-layer waves that propagate toward the ice shelf, temporarily cooling the water by 1-2 °C at 586 m depth over O(10) day timescales (Steiger et al., 2021). While our events exhibit warming between 400 and 600 m depth, this mechanism contradicts our observations, ruling out these waves as the driving force behind the observed features. However, non-local Ekman downwelling may have contributed to the increased advection of WW between July 2021 and January 2023, during which the upper half of the water column beneath TEIS cooled by 1.2 °C to similar depths (Fig. 9c).





## 4.5 Implications

Our results highlight the oceanographic variability beneath TEIS which are related to the need for improved basal melt parameterizations in coupled ice-ocean models. The observed events consistently advect through the water column at around 600 m depth (Figs. 8 and 9), increasing water temperatures between 400 to 600 m depth and potentially enhancing basal melting in regions where ice thickness reaches similar depths, such as along the deep grounding lines of Pine Island and 590  Thwaites Glaciers. By lifting isopycnals closer to the ice-shelf base, these events contribute to localized warming beneath the ice-shelf base and they may accelerate basal melt, with near surface layers potentially continuing to warm in the weeks following an event (Fig. 8a).

Simple depth-dependent melt parameterizations often overestimate heat and salt exchange at the ice-ocean interface, leading to unrealistic projections of grounding-line retreat (Seroussi et al., 2017), and would miss the dynamic events 595  described here. While contemporary models, such as De Rydt et al. (2024), offer valuable insights, they do not yet incorporate ocean surface fluxes, leaving out key processes like polynya activity and sea ice formation, which influence circulation and water mass movements at depth. Since sea ice formation plays a crucial role in redistributing heat, salt, and momentum, its impact on basal melt rates beneath neighbouring ice shelves and the deep grounding lines must be accounted for. Our findings emphasize the importance of incorporating oceanographic processes that link evolving ocean conditions to 600  ice-sheet melting (Yu et al., 2018). Observational data, such as the data presented in this study, provide essential constraints for refining coupled ice-ocean models and improving projections of Thwaites Glacier's future evolution and the potential collapse of WAIS.

## 5 Conclusion

Our measurements revealed coupled atmosphere-ice-ocean interactions that could only be captured using the AMIGOS-3 605  system, which was designed to track long-term water mass movements throughout the water column as PIB sea ice coverage evolved. We observed distinct events occurring in tandem with open ocean conditions or during mobile sea ice cover, where mid-depth waters warm while waters near the seabed temporarily cool over a few weeks. Under a closed fast ice cover in PIB, these events disappear, allowing deep water from Thwaites Trough to penetrate beneath the TEIS. This water mass competes with warmer waters from PIB, which extend far westward reaching beneath TEIS. However, when the fast ice 610  edge retreats across PIB, these competing water masses diminish at depth and upper level waters cool substantially through the increased advection of WW. This highly dynamic system likely influences the basal melting of Thwaites Glacier and other glaciers draining into the Amundsen Sea.

The recent decline in Antarctic sea ice, marked by more extreme annual fluctuations, suggests that the events we observed may become more frequent as sea ice coverage continues to decrease. Reduced sea ice will not only provide less 615  insulation from atmospheric variability but may also allow atmospheric forcing to penetrate even deeper into the water column than previously recognized, influencing the variability of mCDW near the seabed.



## Appendix A: Comparison of in-situ weather data with ERA5 reanalysis

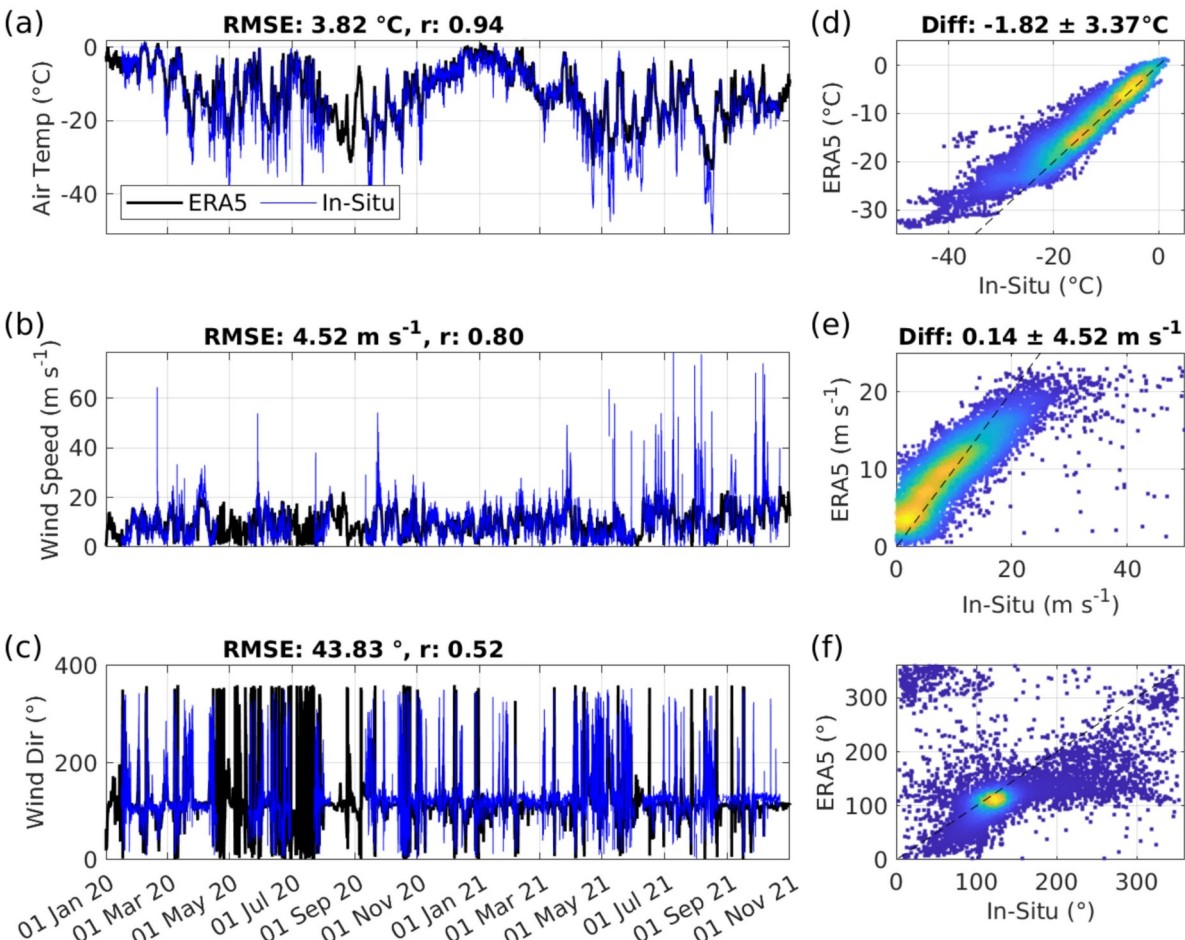

**Figure A1:** AMIGOS-3 versus ERA5: (a-c) Time series of air temperature, wind speed, and wind direction showing available in-situ data. (d-f) Scatter plots showing the relationship between the in-situ and ERA5 data for each variable, with colours indicating point density, where warmer colours correspond to higher point density. Mean differences and their standard deviations are calculated as in-situ data minus ERA5.





## Appendix B: Proxy salinity and density profiles

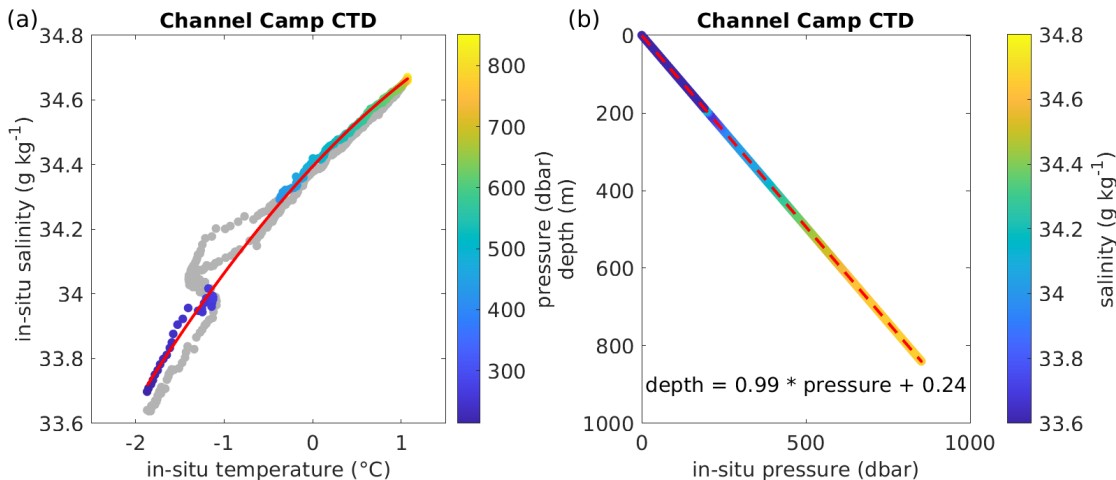

**Figure B1:** CTD cast at Channel Camp. (a) Relationship between in-situ temperature and salinity from CTD profiling on January 12, 2020. Colored dots indicate the data points used to derive a polynomial fit (red curve), excluding the thermocline to reflect long-term averages. (b) Relationship between in-situ pressure and depth below the ocean surface, with the linear fit shown as a dashed red line. Note the transition from freshwater in the borehole to saltwater in the ocean cavity around 200 m depth.


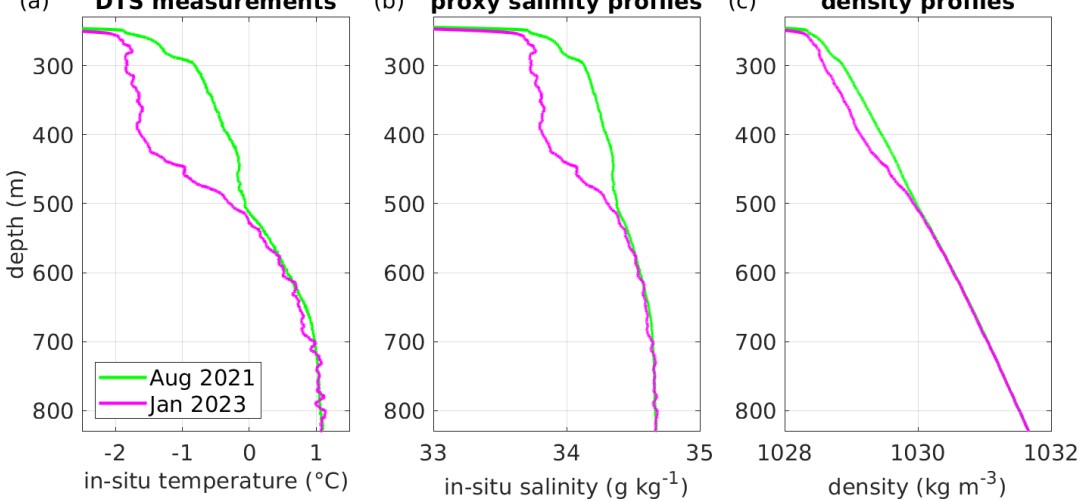

**Figure B2:** Cooling beneath Channel Camp. (a) In-situ temperature profiles. (b) In-situ salinity profiles derived from the polynomial relationship established during CTD profiling on January 12, 2020 (see Fig. B1). (c) Corresponding seawater density profiles. Note the freshening observed in the upper half of the water column.





## Appendix C: Pressure records




**Figure C1:** Pressure records from Channel Camp for (a) shallow levels and (c) near the seabed. Panels (b) and (d) display the continuous wavelet transforms of the two time series. Panel (e) shows the cross-wavelet transform between the two pressure records for their overlapping time period.





**Figure C2:** Pressure records from Cavity Camp for (a) mid-depth levels and (c) near the seabed. Panels (b) and (d) display the continuous wavelet transforms of the two time series. Panel (e) shows the cross-wavelet transform between the two pressure records for their overlapping time period.



**Code availability**

Python code for retrieving daily sea ice concentration can be found at https://github.com/tsnow03/thwaites_amigos.git. The MATLAB Gibbs-SeaWater (GSW) Oceanographic Toolbox is available from http://www.teos-10.org/. MATLAB software for wavelet analysis can be found at https://github.com/grinsted/wavelet-coherence.

**Data availability**

The AMIGOS-3 data are available from the United States Antarctic Program Data Center (USAP-DC) at https://www.usap-dc.org/view/project/p0010162. Borehole CTD and DTS data from Cavity Camp and Channel Camp will be available from USAP-DC upon acceptance of this article. The ship-based CTD dataset is available at https://www.usap-dc.org/view/dataset/601785. Autonomous underwater vehicle data are available at https://snd.gu.se/en (https://doi.org/10.5878/yw26-vc65). ERA5 reanalysis data are available from https://cds.climate.copernicus.eu/datasets/reanalysis-era5-single-levels?tab=overview. Sentinel-1 imagery is available from the Copernicus Open Access Hub (https://scihub.copernicus.eu/). The sea ice concentration dataset is available from the University of Bremen, at https://data.seaice.uni-bremen.de/amsr2/asi_daygrid_swath/s3125/.

**Author contributions**

CTW conceived the study, led data analysis, produced the figures and drafted the manuscript. TS provided sea ice concentration time series and contributed to writing. TSD contributed to the discussion of hydrographic properties. SWT calibrated the DTS data and processed the MicroCAT data. TAS developed the AMIGOS-3 system. ECP led field work, collected and processed borehole CTD casts. ECP and KJH are principal investigators of the TARSAN project. All authors discussed results, implications, edited text and approved of the final manuscript.

**Competing interests**

Karen J. Heywood serves as Co-Editor-in-Chief of *Ocean Science*.

**Acknowledgements**

This research is from the TARSAN project, a component of the International Thwaites Glacier Collaboration (ITGC). We thank Bruce Wallin, Dale Pomraning, Christopher Kratt, Gabriela Collao-Barrios and all members of the TARSAN team.



We would also like to acknowledge the invaluable support from the work centers at McMurdo Station, the WAIS Divide
staff, and Kenn Borek Air throughout event C445. Special thanks to Troy Juniel, Jenny Cunningham and Dean Einersen for
their unwavering commitment during the challenging COVID season in 2021/22. Logistics provided by NSF-U.S. Antarctic
Program and NERC-British Antarctic Survey. ITGC Contribution No. ITGC-144.

**Funding**

This research has been supported by the National Science Foundation, Directorate for Geosciences (grant no. 1929991), and
the Natural Environment Research Council, British Antarctic Survey (grant no. NE/S006419/1). TSD was also supported by
UK Natural Environment Research Council National Capability programme AtlantiS (NE/Y005589/1).

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

**Supplementary materials**

The supplementary material for this article can be found at [link].