# Peer review of "Thwaites Eastern Ice Shelf Cavity Observations Reveal Multi-vear"

_EGUsphere, 2025_

## Author Comment (AC1)

**Response to Reviewer Feedback on Wild et al., 2025**

**"Thwaites Eastern Ice Shelf Cavity Observations"**

**Reviewer 1:**

RV1-1: This manuscript presents ice shelf cavity observations in Pine Island Bay. The value of these observations is indisputable, and they deserve to be published as proposed in this manuscript. I do not have any major criticism. If any, I wish the presentation should be improved to better highlight the novelty of the results, and to provide a slightly improved description of methodologies.

Thank you for your positive feedback and support for publication. We appreciate your suggestions and have enhanced the presentation of the novelty in the revised manuscript. The main changes include: (1) a rephrased abstract (see RV1-2) and parts of the introduction, (2) a more detailed explanation of the wavelet analysis, and (3) rewording of the section on thermal expansion of the water column for the reasons you provided.

RV1-2: Novelty: Reading the abstract only, I find it hard to appreciate what makes this study novel and unique. Could this be restated and improved?

We revised the abstract and parts of the introduction (related RV2-3).

Revised abstract: Pine Island Bay (PIB), situated in the Amundsen Sea, is renowned for its retreating ice shelves and highly variable sea ice. While brine rejection from sea ice formation and glacial meltwater influence seawater properties, the downstream impacts beneath the region's floating ice shelves remain poorly understood. Here, we exploit an unprecedented, multi-year (2020–2023) oceanographic time series from instruments deployed through boreholes beneath the Thwaites Eastern Ice Shelf (TEIS), immediately downstream of PIB, offering new insight into how ice-ocean-atmosphere interactions in PIB shape oceanographic conditions within the subshelf cavity. Our observations reveal a sustained warming and thickening of the modified Circumpolar Deep Water (mCDW) layer near the seabed since January 2020, critical in a region where mCDW drives basal melting beneath West Antarctica's most vulnerable outlet glaciers. Concurrently, the retreat of the multi-year sea ice edge by over 150 km across most of PIB has enhanced the advection of Winter Water, contributing to a cooling of more than 1°C in the upper 250 m beneath TEIS between July 2021 and January 2023. Superimposed on these trends are episodic temperature and salinity anomalies lasting several weeks, originating in PIB and advecting past the mooring. These events link mobile sea ice cover to subshelf hydrography, as mid-depth waters temporarily warm and increase in salinity, leading to an increase in density, while deeper mCDW simultaneously cools and freshens, reducing its density. Overall, these changes are associated with reduced stratification in the cavity. As sea ice continues to decline in a warming Antarctic climate, our results offer a glimpse into how ocean circulation and basal melting may evolve across the Amundsen Sea Embayment. This dataset provides a critical benchmark for refining process-based models and improving melt-rate parametrizations in coupled ice-ocean simulations.

RV1-3: The map in Fig. 1 was so little that I found it hard to read. The choice of colors could also be improved.

[Figure]

Revised map of the study area (Figure 1). Rearranged and with improved colourmap.

RV1-4: DTS thermal profiling: I could not find a definition of the acronym DTS before l. 144. It would be nice to find information about the accuracy of these measurements. Is there a way to assess the potential for temporal drifts?

We have added the acronym to the caption of Fig. 1, and it is defined in the last paragraph of the introduction. Details on the accuracy and spatial resolution of the DTS measurements are included in Sect. 2.1.4, along with a new sentence addressing temporal drifts.

[Figure]

To address the issue of possible drifts, we show here a comparison of temperature measurements from SeaBird and DTS sensors at Channel Camp at the same time and depth, taken at (left) shallow and (right) deeper locations. We assume that the Seabird temperature sensors do not exhibit drift. The DTS data were calibrated using these SeaBird measurements, effectively correcting for temporal drift in the DTS. This calibration also affects the DTS throughout the entire water column.

RV1-5: Cross-wavelet analysis: I found the amount of information available on this method insufficient. What is the unit of the quantity derived from the cross-wavelet transform? How to interpret the result? More generally, would it not be useful to see a wavelet transform of the temperature signal alone? Also, the correlation between temperature and salinity implies some level of density compensation. Would it be possible to analyse the density variations directly? It is partially done in Fig. 6 but it could be better highlighted.

We agree with the suggestion and have now included the continuous wavelet transform of the density variations alone in Figure 3, which also necessitated rewording parts of Section 3.1. In Figure 6, we retain the cross wavelet transform to illustrate the covariance between density variations with wind and currents. Additionally, we have expanded the Methods section to provide further details on the wavelet analysis and clarify the interpretation of the results.

[Figure]

Revised wavelet Figure 3, now showing the continuous wavelet transform of density and labels for cross-referencing between Figures with a perceptually uniform colourmap.

RV1-6: l. 390: I do not understand the statement that warming leads to thermal expansion of the water column. The direct effect of thermal expansion on the position of an isopycnal would be at a centimetric scale at best, especially in such a cold region. This is not something I expect can be directly observed. Can you clarify?

We agree with the reviewer's point and have reworded to clarify that isopycnal displacements due to thermal expansion are minimal at mid-depth and deep layers, consistent with the expected centimetric-scale effect in this cold environment.

RV1-7: Fig. 8: I find it very hard to understand what the x-axis corresponds to and how to read this figure. More details would help.

The x-axis in Fig. 8 represents the estimated horizontal length scale of the advecting features. This scale is calculated by combining daily averaged DTS temperature anomalies with daily averaged Aquadopp current measurements. We assume that the average of the mid-depth currents (521 m) at Cavity Camp and the near-bottom currents (784 m) at Channel Camp approximates the mean current speed throughout the water column. While this assumption is a simplification, it reflects the best estimate possible given the available data. To improve clarity, we have revised the figure description in the main text and updated the x-axis label accordingly.

[Figure]

Revised Figure 8 with a new x-axis label, showing features located between (a) 5-40 km, (b) 5–24 km, and (c) 10–140 km. The approximate feature length scales are given by the range within each interval.

---

## Author Comment (AC2)

**Response to Reviewer Feedback on Wild et al., 2025**

**"Thwaites Eastern Ice Shelf Cavity Observations"**

**Reviewer 2:**

RV2-1: The richness of the datasets and material presented in the Figures here is appreciated, however one easily loses track of the correlations in time between T, S, sea ice and current data, and of the bigger picture conclusions, especially if one is not familiar with the region.I suggest the following:

We thank the reviewer for the positive feedback and thoughtful comments. We have incorporated their suggestions to enhance the clarity of our big-picture conclusions and to better describe the correlations between environmental drivers and observed variability in the subshelf cavity.

RV2-2a: 1) the figures and their content could be introduced more properly (often times a relatively complex result is stated and then just "Fig.x" in brackets, and the reader has to go to the figure first, read the caption and then try to make sense of it)

We have revised the Results section to better introduce the figures and their content. Specifically, we now include introductory sentences that explain the interpretation and purpose of the continuous wavelet transforms, Θ-SA diagrams, feather plots and cross-wavelet transforms. These additions clarify how each method is used to identify periods of changing oscillatory behavior, distinguish between water masses, and detect mixing processes.

RV2-2b: 2) introduce more subsections with goal of the subsection and make a "big-picture summary" at the end of each subsection.

We added new subsections in the Results, each with a clear goal and a summary statement. These include:
- Linking subshelf cavity observations to PIB-sourced waters
- Tracing glacial meltwater and Winter Water mixing beneath TEIS
- Linking environmental drivers and density variations across depths
- Consistent thermal patterns observed during events

RV2-2c: 3) mark relevant times in the Fig. so one can keep better track of what is happening when in different variables

Periods of elevated current speeds are already indicated as grey background shading in Figures 2 and 6. Additionally, we have labeled the key events directly in Figure 3 to aid cross-referencing between figures and variables. Related RV1-5 and RV2-11.

RV2-2d: 4) a summary sketch at the end summarizing bigger picture events

[Figure]

Revised Fig. 11 with additional labels and more precise language.

RV2-2e: 5) a better and bigger map at the beginning to show what is relevant.
We have increased the map sizes in the panels of Figure 1 (related RV1-3).

RV2-3: In the introduction, can you specify more clearly what is different from previous work and which additional information you will analyze here.

We revised the penultimate paragraph of the introduction to more clearly distinguish our study from previous work. The updated text emphasizes the extended observational period, the improved vertical and temporal resolution of our dataset, and our focus on the role of different upstream sea ice types in modulating ocean conditions beneath TEIS.

RV2-4: mark in Fig.2 relevant time periods that are emphasized in the text to be able to keep track

This is a good suggestion and we have marked these as shown below in the revised Figure 2.

[Figure]

Revised Figure 2 with labels to better cross-reference with the main text

RV2-5: solid line is not the mean

We agree and clarify that the figure shows only anomalies of temperature (blue) and salinity (red), not mean values. Although the existing caption already states this clearly, we also changed how the colors are represented in the legend.

RV2-6: line 228: at depth, the warming trend starts more like in August 2020, before that there is a cooling trend?

At depth, warming begins around April 2020, though superimposed events can obscure this. We clarified this in Section 3.1. and adjusted Fig. 2 with additional labels.

RV2-7: line 240 : I don't really see the warming trend at depth from April on, more from August 2020 on?
We agree that temperatures at both deep layers decrease from January to April 2020. After this period, a clear warming trend is evident in both records, though temporarily obscured by superimposed event-related cooling anomalies (Fig. 2c/d).

RV2-8: line 262: from Fig. 3 it is not so clear that co-variation happens at all periods. Even though the arrows show co-variation the significance clearly drops sharply with small periods.

We revised the figure to show continuous wavelet transforms of density instead of cross-wavelet transforms between temperature and salinity (related RV1-5). We agree that statistical significance decreases sharply for periods shorter than 0.5 days. The reworded text now emphasizes density variability and highlights the clear tidal signals visible as fortnightly power bands around 0.5 and 1 day periods.

RV2-9: just visually from Fig. 2, co-variance at long time scales is not evident for the Channel Camp 316m, there is a clear warming but no freshening trend.

This comment is now obsolete due to the shift from temperature/salinity co-variance analysis to density variability alone, which better fits to the narrative and aligns with the subsequent comparison to environmental drivers, as suggested by RV1-5. Additionally, the April 2020 event is now clearly visible at Channel Camp (see revised Fig. 3a).

RV2-10: Fig.3, Fig.6 add colorbar
Added colourbars and descriptive text to the methods (related RV1-5).

[Figure]

Revised Figure 6 with colourbar and perceptually uniform colourmap.

RV2-11: line 277: to be able to keep better track of the distinct events, somehow marking them in Fig.3 would be good, and relate them back to the events/trends evident from Fig.2

The grey shading in the background of Figs. 2 and 6 was already included to indicate time periods of elevated current speeds and facilitate cross-referencing of events across figures. To further improve clarity and help track distinct events, we now explicitly label the events and key findings in Fig. 3, linking them more clearly to trends evident in Figs. 2 and 6.

RV2-12: line 289: it seems that something new is discussed now, so a new subsection would help, together with a brief intro on what Fig.4 shows to help the reader digest the information.

We have introduced several new subsections in the Results section, including brief introductions to the Figures (related RV2-2b)

RV2-13: line 308: was the "Gade line" introduced before?
The Gade line is introduced here. We have now added an introductory sentence that links the T/S diagrams to mixing lines.

RV2-14: line 353: something important and new is starting here, so a subsection would be good. I like the introductory sentence as motivation, something like this should appear at the beginning of each new subsection.

We have added the subsection titled 'Linking environmental drivers and density variations across depths'

RV2-15: line 367: something new starts again

We have added the subsection titled 'Consistent thermal patterns observed during events'

RV2-16: Fig.7 would it make sense to somehow plot differences in time or wrt the mean to emphasize trends, the lines are all very close together, mark relevant depth intervals mentioned in the text?

In addition to temperature values in Fig. 7, we also plot the anomalies relative to the first profile in each event, which are later used to determine the horizontal length scale as presented in Fig. 8. To improve clarity, we have added markers for the relevant depth intervals discussed in the text, including the transition depth between warming and cooling layers.

[Figure]

Revised Figure 7 with depth intervals, transition point and labels. Bottom row shows anomalies from the first profile in the time span.

RV2-17: is Fig.10 i,j,k discussed?

Thank you for noting this. We corrected the typos in the figure references within the section 'Sea ice conditions in PIB: formation and breakup of fast ice'.

RV2-18: Fig.11: I appreciate it, since it shows a big-picture schematic summary, but please properly take the reader through Fig.11

We added several more precise labels to this summary sketch (see RV2-2d)

RV2-19: it would be great if at the end there could be a summary sketch, summarizing the major findings and conclusions that were evident in all the figures thereby bringing them together

Redundant. See reply to RV2-18

**Editor**:

E-1: consider swapping out the rainbow/jet colour scheme for something else in due course

Changed all cross-wavelet transforms to a perceptually uniform colourmap (related RV2-10).

**Channel Camp**

[Figure]

[Figure]

E-2: data statement to update

We updated the data availability statement.

E-3: update acknowledgements

We added a sentence acknowledging the editors and referees.